# Position: Reframing Hallucination: Latent Space Geodesics as a Pathway for Generative Discovery

**Zhihao Hao** [1]   **Bob Zhang** [2]   **Haisheng Li** [1]

## Abstract

Although evaluation practice for generative models has moved beyond purely retrieval-based metrics, many protocols still penalize deviations from known outputs, limiting their use in scientific discovery and creative reasoning. This position paper argues that uniformly suppressing such deviations can induce epistemic mode collapse, causing models to favor safe reproduction over exploratory hypothesis generation. We propose the Higher-Dimensional Cognitive Hypothesis (HDCH), which interprets some valuable hallucinations as high-dimensional latent-space traversals that appear erroneous when projected onto established knowledge. We distinguish Type I outputs, which are factually inconsistent or structurally incoherent, from Type II exploratory hypotheses, which are novel, structurally coherent, and worth further validation. Through controlled demonstrations, we illustrate that discovery-oriented generation benefits from calibrated instability, with exploratory yield peaking near a critical transition regime rather than increasing monotonically with randomness. We further advocate an Exploratory Signal-to-Noise Ratio (ESNR) framework that combines distributional divergence with external structural validation, shifting evaluation from static retrieval validation toward calibrated latent exploration.

---

[1]School of Computer Science and Artificial Intelligence & Beijing Key Laboratory of Applied Statistics and Digital Regulation & Beijing Key Lab of Commercial Data Security Protection and Intelligent Governance & Academy for Interdisciplinary Studies, Beijing Technology and Business University, Beijing 100048, China [2]Department of Computer and Information Science, Faculty of Science and Technology & Centre for Artificial Intelligence and Robotics, Institute of Collaborative Innovation, University of Macau, Macau 999078. Correspondence to: Zhihao Hao <zhihao.hao@btbu.edu.cn>, Haisheng Li <lihsh@btbu.edu.cn>.

*Proceedings of the 43rd International Conference on Machine Learning*, Seoul, South Korea. PMLR 306, 2026. Copyright 2026 by the author(s).

## 1. Introduction

The trajectory of generative artificial intelligence has arrived at a critical juncture, constrained in part by evaluation practices inherited from discriminative and retrieval-oriented modeling. Traditional metrics such as BLEU, ROUGE, and exact match remain useful for tasks with well-defined references, but they penalize deviation from known outputs and therefore provide a limited view of discovery-oriented generation. Recent evaluation practice has begun to move beyond purely retrieval-centric metrics, incorporating diversity measures, open-ended preference benchmarks, dynamic environments, and LLM-as-a-Judge protocols. However, these approaches still lack a unified account of when deviation should be treated as incoherent error and when it should be preserved as a structurally coherent exploratory hypothesis. While strict factuality is essential for applications such as medical diagnosis, applying the same zero-deviation standard to open-ended discovery tasks, such as creative reasoning and hypothesis generation, constitutes a category error. This creates a bottleneck that penalizes exploratory behavior, inducing a form of mode collapse where models converge on safe, repetitive outputs (Atienza, 2025). Consequently, we face a fundamental misalignment: optimizing solely for retrieval agreement or low perplexity can suppress the generation of novel insights. **This position paper argues that hallucinations should not be universally minimized, but rather reinterpreted through a geometric lens as potential mechanisms for latent-space exploration and scientific discovery.**

We propose that resolving this misalignment requires a geometric perspective on how generative models explore their latent spaces. These models operate in hundreds or thousands of dimensions, far beyond human intuition. According to the Manifold Hypothesis (Meilă & Zhang, 2024), training data occupies a low-dimensional manifold within this vast space (Sharma & Kaplan, 2022). We posit that novel, valuable outputs often arise when a model traverses geodesics (the shortest paths) in the high-dimensional ambient space, venturing outside the known manifold. When projected back onto our low-dimensional manifold of established knowledge, these trajectories appear as discontinuities and are labeled as "hallucinations." This is not merely a statistical

error but often a perceptual artifact of dimensional compression. Projects such as DeepMind's GNoME suggest that generative exploration can propose candidates absent from existing databases, some of which may later be validated by domain-specific physical constraints (Merchant et al., 2023). Similar discovery-oriented pressures arise in food computing and sensory intelligence, where models must reason over multimodal, spatial-temporal, olfactory, and gustatory signals rather than merely retrieve known labels or descriptors (Hao et al., 2025d;b).

The core challenge, therefore, is the lack of metrics to discriminate between two fundamentally different phenomena: harmful confabulations (noise) and useful exploratory deviations. We formalize this as the distinction between Type I errors (factually inconsistent, structurally incoherent noise) and Type II exploratory hypotheses (factually novel yet structurally coherent propositions about unobserved regions of the data distribution). Current paradigms, including safety-focused techniques like Retrieval-Augmented Generation (RAG) (Walker et al., 2025), inadvertently suppress Type II deviations by tethering generation to existing documents, thereby stifling the combinatorial creativity essential for discovery.

This position paper formalizes this distinction and proposes a new evaluation paradigm centered on calibrated exploration. Our primary contributions are:

1. **The Higher-Dimensional Cognitive Hypothesis (HDCH):** We propose a geometric interpretation of selected generative deviations as latent-space traversals that may function as discovery pathways, rather than treating all deviations as statistical failures.

2. **A Type I/Type II Error Taxonomy:** We distinguish factually inconsistent and structurally incoherent outputs from factually novel but structurally coherent exploratory hypotheses, providing a conceptual basis for evaluating generative novelty.

3. **The Exploratory Signal-to-Noise Ratio (ESNR):** We introduce an evaluative criterion that balances distributional divergence against structural coherence, offering a direction for discovery-oriented evaluation beyond retrieval accuracy.

The remainder of this paper is organized as follows. Section 2 details the Higher-Dimensional Cognitive Hypothesis and the geometry of latent exploration. Section 3 operationalizes the Type I/II taxonomy, defines the ESNR metric, and introduces the thermodynamic Safety Sandbox. Section 4 presents controlled demonstrations on latent traversal, symbolic filtering, and phase-transition behavior. Section 5 addresses theoretical counterarguments regarding safety and falsifiability. Section 6 discusses the risks of epistemic

mode collapse and proposes distributional robustness protocols. We conclude in Section 7 with a roadmap for evolving community benchmarks.

**Conflict of Interest Disclosure.** The authors declare no financial conflicts of interest related to this work.

## 2. The Higher-Dimensional Cognitive Hypothesis

The Higher-Dimensional Cognitive Hypothesis (HDCH) posits that specific generative deviations, typically categorized as hallucinations, represent projections of valid geodesic traversals in a high-dimensional latent space $\mathcal{Z}$. These traversals extend beyond the constraints of the human epistemic manifold $\mathcal{M}$. As illustrated in Figure 1, what appears as a discontinuous "jump" or error in the low-dimensional observation space is often a smooth, continuous path in the high-dimensional reality. This section formalizes the geometric mechanisms underlying these traversals and discusses motivating cases suggesting that some off-manifold excursions may support novelty in scientific and abstract reasoning.

### 2.1. Geometry of Latent Traversal

Generative models, such as Transformer-based architectures, approximate a data distribution $P(X)$ supported on a low-dimensional manifold embedded in a high-dimensional ambient space $\mathbb{R}^D$. While human cognition is intuitively anchored in low-dimensional perceptual spaces (e.g., 3D space + time), models like GPT-4 operate in latent spaces where $D$ is on the order of hundreds or thousands ($D \gg 512$).

Classic representation learning theory suggests that semantic relationships are preserved as linear directions in this latent space (Mikolov et al., 2013). We argue that "hallucinations" often occur when the model infers a connection between two distant concepts $A$ and $B$ via a geodesic path $\gamma_{AB}$ that does not lie on the training manifold. While $\gamma_{AB}$ minimizes the energy function in $\mathbb{R}^D$, its projection $\pi(\gamma_{AB})$ onto the lower-dimensional manifold of established facts $\mathcal{M}$ appears discontinuous. Attention mechanisms, by capturing long-range dependencies (Vaswani et al., 2017), effectively enable the model to form direct, high-dimensional associations between semantically distant concepts—associations that may appear as discontinuous "jumps" when projected onto a lower-dimensional, locally continuous human conceptual map.

### 2.2. Cognitive Validity and the Projection Metric

To distinguish between stochastic noise (Type I error) and valuable geodesic traversal (Type II exploration), we introduce the Cognitive Validity Metric $\Psi(\mathbf{z})$. Let $\pi : \mathcal{Z} \to \mathcal{M}$

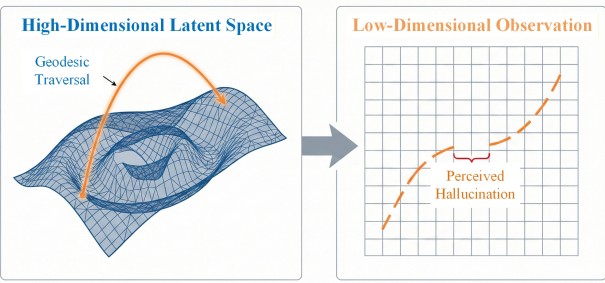

*Figure 1.* The Geometry of Hallucination under the HDCH Framework. (a) High-Dimensional Reality ($\mathcal{Z}$): The model traverses a continuous geodesic (orange trajectory) on a complex manifold (blue spiral), venturing off the training distribution to connect distinct semantic regions. (b) Low-Dimensional Observation ($\mathcal{M}$): When projected onto the human epistemic plane, the hidden dimension is lost. The continuous trajectory appears as a discontinuous jump or gap, which accuracy-centric metrics misclassify as a "hallucination" rather than a valid bridge.

be a smooth mapping from the latent space to the interpretable output space. We define $\Psi$ as a function of the local geometric distortion and the semantic alignment:

$$\Psi(\mathbf{z}) = \underbrace{\left( \frac{\|J_\pi(\mathbf{z})\|_F}{\|J_\pi(\mathbf{z}_0)\|_2} \right)}_{\text{Local Distortion}} \cdot \underbrace{\exp\left( -\frac{D_{KL}(p_{\text{proj}}\|p_{\text{fact}})}{\beta} \right)}_{\text{Semantic Alignment}} \quad (1)$$

Where:

- $J_\pi(\mathbf{z})$ is the Jacobian matrix of the projection $\pi$ at latent point $\mathbf{z}$. The Local Distortion term quantifies the complexity and novelty of the information synthesis at point $\mathbf{z}$. A high ratio of the Frobenius norm implies that the projection $\pi$ is highly "twisted" or expanded in this region, corresponding to the process of fusing multiple independent conceptual features into a novel output structure.

- $D_{KL}$ is the Kullback-Leibler divergence between the projected output distribution $p_{\text{proj}}$ and the factual baseline $p_{\text{fact}}$.

- $\beta \in (0, +\infty)$ is the Exploration Temperature. When $\beta \to 0$, the penalty for divergence is infinite (strict retrieval). When $\beta > 1$, the system enters a high-temperature regime favoring entropy maximization, allowing the model to traverse regions of the latent space that are structurally valid but factually unmapped.

This formulation connects information geometry with the cognitive theory of Conceptual Integration Networks (Fauconnier & Turner, 2008). A high Local Distortion term implies a "double-scope blend," where the model synthesizes disparate frames into a novel structure. The metric

$\Psi(\mathbf{z})$ thus rewards outputs that are structurally complex even if they diverge significantly from established facts, provided $\beta$ is tuned to permit exploration.

### 2.3. Illustrative Cases of Discovery-Oriented Generation

Several recent examples suggest that outputs initially viewed as deviations from known data can later become valuable candidates for verification. We use these cases as motivating examples rather than as direct evidence that all hallucinations are beneficial.

**Materials Science (GNoME).** DeepMind's GNoME project used a large-scale generative and screening pipeline to propose candidate crystal structures (Merchant et al., 2023). Many of these candidates were absent from existing crystal databases and therefore would be penalized by purely retrieval-based evaluation. However, subsequent Density Functional Theory (DFT) calculations identified a substantial subset as thermodynamically stable. This case illustrates the distinction between factual absence and structural incoherence: a candidate may be novel with respect to the training record while still satisfying domain-specific physical constraints. In our terminology, such outputs motivate the notion of Type II exploratory hypotheses.

**Structural Biology (AlphaFold).** Protein-structure prediction provides another example where a mismatch with a static reference structure does not always imply a meaningless error. Models such as AlphaFold generate structural hypotheses that are evaluated against experimental structures, but biological systems may exhibit conformational variability, disorder, or context-dependent dynamics (Jumper et al., 2021). This does not mean that every disagreement with a reference structure is valuable. Rather, it suggests that some deviations should be examined through additional structural or functional constraints before being discarded as noise.

**Cross-Domain Conceptual Analogy.** Generative models can also produce analogies across distant conceptual domains. Many such analogies are weak or misleading, but some may reveal useful structural correspondences that support hypothesis formation or theoretical reframing. We treat these outputs as heuristic candidates, not as verified discoveries. Their value depends on whether subsequent analysis can establish coherence, explanatory utility, and compatibility with domain constraints.

Together, these cases motivate a broader evaluation question: when should a generative deviation be rejected as a Type I error, and when should it be preserved as a Type II exploratory candidate for further validation?

### 2.4. Connection to Existing Frameworks

The HDCH complements the World Model theory (Ding et al., 2025; Bar et al., 2025), which posits that agents learn

a compressed spatial-temporal representation of their environment. While World Models focus on predictive accuracy for control, HDCH extends this to abductive generation. This form of generation aims to produce new, testable hypotheses that extend the boundary of knowledge, rather than merely explaining existing data. Thus, an evaluation framework must optimize for the Exploratory Signal-to-Noise Ratio (ESNR) rather than simple likelihood, distinguishing between errors that degrade the model and deviations that expand it.

HDCH is also related to isomerism learning, which studies how heterogeneous models can collaborate without being forced into a single homogeneous architecture (Hao et al., 2023a). This perspective supports one design principle of our framework: exploratory candidates should not be evaluated only by the same model distribution that generated them, but should be exposed to structurally different forms of validation.

# 3. Metrics for Latent Exploration and Safety

Current evaluation frameworks prioritize factual correctness and impose strict penalties on deviation. This creates an imbalance that hinders the assessment of the exploratory potential inherent in generative systems. To address this, we propose a thermodynamic evaluation framework. This framework integrates factual reliability with creative reasoning using metrics designed to assess the geometry of latent-space exploration.

## 3.1. The Stagnation of Accuracy-Centric Evaluation

Prevailing benchmarks such as Exact Match or ROUGE measure adherence to the training distribution. They operate under the assumption that the goal of generation is retrieval. This imposes a manifold constraint. It forces the model to collapse its high-dimensional cognitive activity onto a low-dimensional surface of known facts.

This constraint manifests as a penalty for novelty. Metrics often classify outputs as errors even when they represent valid scientific hypotheses in unmapped regions of the solution space. Analyses of gradient magnitudes suggest that shallow traversals link to high factual accuracy but low diversity, whereas deep traversals may uncover less conventional but potentially valuable patterns. This view is consistent with neuroscience evidence linking creative cognition to the default mode network and its dynamic interaction with executive-control systems (Shofty et al., 2022; Chen et al., 2025; Liu et al., 2024). Current benchmarks fail to capture this distinction between incoherent deviation and potentially useful exploration.

## 3.2. The Latent-Traversal Score

To quantify the magnitude of exploration, we define the Latent-Traversal Score, denoted as $S_t$. Unlike trajectory-based integrals which can be computationally ambiguous, we define $S_t$ for a generated output with latent representation $\mathbf{z}$ as its Euclidean deviation from a reference point $\mathbf{z}_{ref}$ (typically the centroid of the training distribution in $\mathcal{Z}$).

$$S_t(\mathbf{z}) = \|\mathbf{z} - \mathbf{z}_{ref}\|_2 \qquad (2)$$

Alternatively, to capture the energy required for such a traversal, one could measure the norm of the gradient of the task-specific loss at $\mathbf{z}$ relative to the reference. A high $S_t$ indicates the model has ventured into a region of high semantic energy or rapid conceptual change, analogous to crossing a phase boundary.

We hypothesize a logarithmic relationship between the Latent-Traversal Score $S_t(\mathbf{z})$ and the Cognitive Validity metric $\Psi(\mathbf{z})$ defined in the previous section. This aligns with the Weber-Fechner law of perception:

$$S_t(\mathbf{z}) = \alpha \log[\Psi(\mathbf{z})] + \epsilon \qquad (3)$$

The parameter $\alpha$ serves as a calibration factor. The term $\epsilon$ captures stochastic exploratory noise. This hypothesis suggests that the marginal gain in discovery diminishes as the model ventures further from established knowledge.

## 3.3. Exploratory Signal-to-Noise Ratio (ESNR)

A high traversal score is necessary but not sufficient for discovery. We must distinguish useful exploration from incoherent noise. We introduce the Exploratory Signal-to-Noise Ratio to filter Type I errors from Type II exploratory candidates.

$$\text{ESNR}(x, \mathbf{z}) = \frac{C_{\text{ext}}(x; \mathbf{z})}{D_{KL}(p_{\text{proj}} \| p_{\text{fact}}) + \delta}, \qquad (4)$$

where $C_{\text{ext}}(x; \mathbf{z})$ denotes structural coherence evaluated by a validator that is decoupled from the generator, and $\delta > 0$ is a small constant used to avoid numerical instability when the divergence is close to zero.

The numerator, Structural Coherence, measures the internal plausibility of the output $x$ under an external or decoupled validator. In domains with explicit rules, this validator can be instantiated as a domain-specific function, such as a chemical valency checker, a code compiler, a symbolic proof verifier, or a physical simulator. This decoupled-evaluation view is related to recent work on dynamic collaboration among heterogeneous models, where structurally different learners provide complementary signals rather than sharing

a single model space (Hao et al., 2025c). In open-ended domains where such validators are unavailable, an instructionally decoupled LLM-as-a-Judge can serve as a heuristic validator of internal consistency and structural plausibility. Importantly, this judge should not be asked to verify empirical truth against its own parametric memory. Its role is limited to assessing whether the output is coherent, self-consistent, and compatible with the stated constraints.

The denominator measures Distributional Divergence, quantifying how far the output deviates from the training distribution. Here, $p_{\text{fact}}$ approximates the training data distribution or a calibrated reference model, while $p_{\text{proj}}$ represents the current model's output distribution conditioned on $\mathbf{z}$.

A high ESNR alone does not define a Type II discovery, because standard retrieval outputs may also have high coherence and low divergence. We therefore identify Type II exploratory candidates by the intersection of high structural coherence and sufficiently high distributional divergence. In contrast, Type I errors are characterized by high divergence but low structural coherence. Thus, ESNR should be viewed as a screening criterion for candidate hypotheses, not as proof of truth. Final validation still requires external grounding through formal verification, simulation, empirical experiments, or delayed domain-specific assessment.

### 3.4. Thermodynamic Safety Sandboxes

We propose a Safety Sandbox architecture to manage high-ESNR outputs. This framework serves as the key engineering implementation for calibrated exploration. It reconciles the tension between safety and innovation by restricting high-risk exploration to controlled environments. Related work on blockchain-powered federated learning suggests that decentralized trust and model-integration mechanisms can support controlled collaboration among heterogeneous participants (Hao et al., 2025e).

Dynamic Isolation creates temporary reasoning environments. These environments allow the model to explore high-temperature states without contaminating the user-facing interface.

The ESNR Gate acts as a semantic firewall. It routes outputs based on the ratio defined in Equation 4. Outputs with low ESNR are discarded as noise. Outputs with high ESNR are preserved and tagged as hypothetical.

Gradient Throttling limits exploration speeds. If the Latent-Traversal Score $S_t$ exceeds a safety threshold defined by $k\sigma$ (e.g., $k = 2.3$, corresponding to a 99% confidence interval under a Gaussian assumption), the system automatically reduces the sampling temperature. This ensures that the model does not diverge into complete incoherence.

This approach operationalizes the HDCH in a safety-aware

setting. It treats hallucinations not as binary failures but as probabilistic candidates for discovery. It ensures that the AI operates within safe boundaries while maximizing its potential for scientific insight.

## 4. Controlled Demonstrations

These experiments are intended as controlled demonstrations of the proposed evaluation principles, rather than as comprehensive benchmarks for deployed generative systems. We present three experiments to illustrate the Higher-Dimensional Cognitive Hypothesis under simplified but interpretable settings. The first experiment operates in a continuous latent space and examines whether exploratory traversal can reach a held-out valid mode. The second experiment operates in a discrete symbolic space and evaluates whether the ESNR criterion can separate novel logical discoveries from incoherent noise. The third experiment studies the non-monotonic relationship between sampling temperature and discovery yield, motivating the need for calibrated exploration in safety-aware generation.

### 4.1. Experiment 1: The Hidden Manifold Task

This experiment examines whether optimizing for the Latent-Traversal Score can help a model reach valid data modes absent from the training distribution.

**Experimental Setup**  We construct a synthetic dataset in a 100-dimensional space consisting of three disjoint manifolds. We term these Islands A, B, and C. Islands A and B constitute the training set and represent established knowledge. Island C is held out and represents a latent scientific discovery. All islands follow the same underlying energy function but are separated by high-energy barriers. We train a Variational Autoencoder on A and B. Structural coherence in this task is computed analytically using the ground-truth energy function of the synthetic manifold. Samples with low energy under this function are treated as coherent candidates, including samples that are absent from the training islands but remain compatible with the underlying energy landscape.

**Visualization Analysis**  Figure 2 visualizes the results. We project the high-dimensional data onto two principal components. The gray contours depict the implicit energy landscape where darker regions indicate lower energy states.

The baseline model utilizes standard low-temperature sampling. Its outputs appear as blue circles in the figure. These samples cluster tightly around Islands A and B. This confirms that strictly minimizing reconstruction error leads to mode collapse. The baseline model fails to cross the energy barriers to find the hidden manifold.

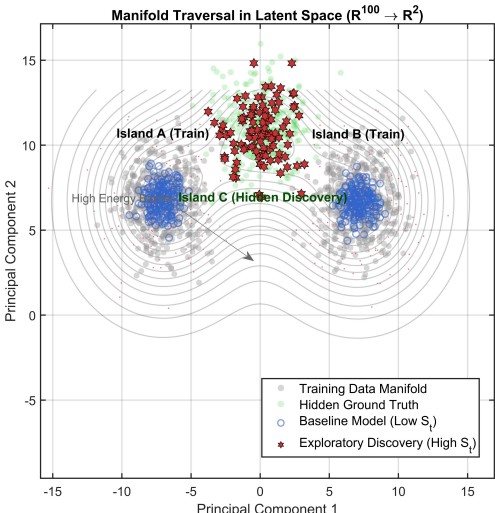

*Figure 2.* Visualization of the Hidden Manifold Task. The plot displays the projection of 100-dimensional latent states onto two principal components. The gray contours represent the energy landscape. The blue points indicate baseline samples trapped in local minima A and B. The red points indicate exploratory samples that successfully traversed the high-energy barrier to discover the hidden Island C.

The experimental model utilizes an exploration-biased strategy. This strategy maximizes the traversal score. Its outputs include the red points in the figure. The visualization shows that the model traverses the high-energy barrier between the training modes. Approximately 12 percent of its trajectories converge to the hidden Island C. These outputs possess low energy under the ground-truth function. This supports interpreting them as candidate discoveries rather than mere errors.

### 4.2. Experiment 2: The Omitted Axiom Task

This experiment simulates scientific hypothesis generation in a symbolic domain. It examines whether the ESNR criterion can distinguish between incoherent hallucination and novel deduction.

**Setup and Protocols** We define a formal axiomatic system with three generating rules. The training dataset contains valid theorems derived exclusively from the first two rules. Theorems derived from the third rule are valid within the system but absent from the training data. We train a Transformer-based language model on this restricted set. We then generate sequences using a high temperature to induce hallucinations. We classify the outputs into retrieval, syntactic noise, and novel discovery categories. Structural coherence is evaluated by an external deterministic symbolic verifier. The verifier checks whether a generated sequence satisfies the rules of the underlying axiomatic system, including the held-out third rule, and is independent of the

generator's likelihood score.

**Results Analysis** Figure 3 presents the distribution of generated outputs plotted against Distributional Divergence (x-axis) and Structural Coherence (y-axis). The plot reveals three distinct clusters:

1. **Retrieval (Blue Circles):** These outputs exhibit low divergence and high coherence. They correspond to known theorems derived from the training axioms.

2. **Type I Noise (Gray Crosses):** These outputs exhibit high divergence but low coherence. They represent syntactic errors and logical discontinuities.

3. **Type II Discovery (Red Diamonds):** These outputs exhibit both high divergence and high coherence. They correspond to valid theorems derived from the hidden third rule.

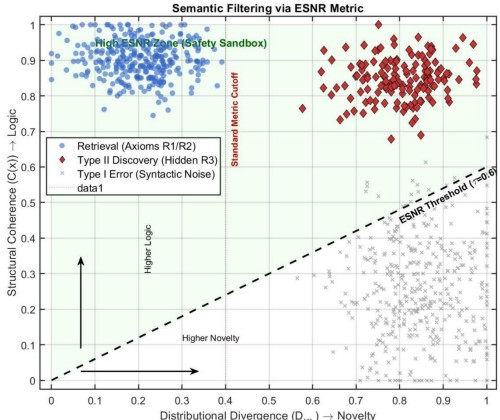

*Figure 3.* Semantic Filtering via the ESNR Metric. The scatter plot positions generated outputs based on their novelty (x-axis) and logical validity (y-axis). Standard metrics (vertical red cutoff) reject high-novelty outputs regardless of validity. The ESNR metric (diagonal black cutoff) successfully distinguishes between Type I syntactic noise (gray crosses) and Type II valid discoveries (red diamonds).

**Metric Comparison** The vertical red dotted line in Figure 3 represents the cutoff for standard accuracy metrics like BLEU. These metrics prioritize low distributional divergence. Consequently, they correctly accept the Retrieval cluster but reject both the Noise and the Discovery clusters. This illustrates how accuracy-centric evaluation suppresses innovation.

In contrast, the diagonal dashed line represents the ESNR threshold. This metric accepts outputs where structural coherence rises in proportion to novelty. The ESNR filter successfully captures the Type II Discovery cluster (red diamonds) while rejecting the Type I Noise cluster (gray

crosses). Quantitative analysis shows that this filter retained 94 percent of the novel theorems while removing 98 percent of the noise.

**Implication**  This suggests that logical consistency can provide a useful proxy for screening candidate hypotheses when direct factual support is unavailable. The result also indicates that the framework can be instantiated in discrete symbolic reasoning settings relevant to Large Language Models.

### 4.3. Experiment 3: Thermodynamic Phase Transitions

This experiment investigates the non-monotonic relationship between sampling temperature and discovery yield. It examines the hypothesis that discovery may concentrate near an "edge-of-chaos" regime, requiring calibrated rather than maximal instability.

**Setup**  We simulate the output distribution of a generative model across a temperature range from 0.1 to 2.5. We model the generative process using a Boltzmann distribution where probability mass is determined by the trade-off between energy and entropy. We define three macrostates: Retrieval, which corresponds to deep but narrow energy wells; Discovery, which corresponds to higher energy but broader metastable states; and Noise, which represents the high-entropy background of the latent space. Structural coherence is scored using a deterministic syntax parser, which separates well-formed exploratory outputs from high-temperature syntactic noise. This allows the experiment to measure not only how far the model moves from the retrieval regime, but also whether the resulting outputs remain structurally admissible.

**Results Analysis**  Figure 4 illustrates the distinct thermodynamic regimes. At low temperatures ($T < 0.7$), the energy term dominates. The system freezes into the local minima of the training data, resulting in pure Retrieval (blue curve). As $T$ approaches a critical threshold ($T \approx 1.0$), the entropic contribution increases. This allows the system to escape the narrow training wells and access the Discovery macrostate. The rate of Type II Discovery (red curve) spikes dramatically, peaking at $T = 1.2$. Beyond this point, as $T$ exceeds 1.5, the massive entropy of the Noise state overwhelms the structural priors. The system disintegrates into incoherence (gray curve).

**Implication**  This unimodal response function cautions against the naive assumption that more randomness necessarily yields more creativity. It suggests that discovery-oriented outputs may concentrate near a critical regime. This motivates the Gradient Throttling mechanism proposed in our Safety Sandbox. The sandbox acts as a thermostat.

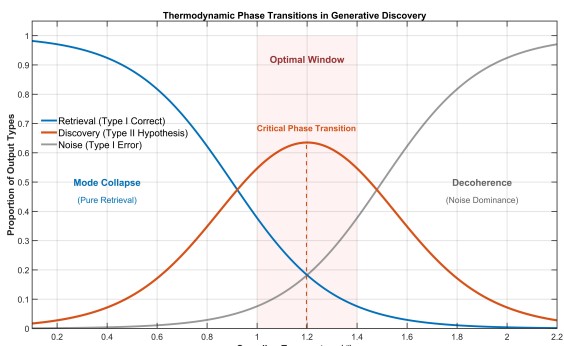

*Figure 4.* Thermodynamic Phase Transitions in Discovery. The graph plots the proportion of output types against sampling temperature. A critical phase transition is observed around $T = 1.2$, representing the optimal window for discovery. Temperatures below this window result in mode collapse (pure retrieval), while temperatures above result in decoherence (noise dominance).

It dynamically adjusts the system temperature to maintain it within the phase transition boundary, effectively balancing the drive for novelty against the constraint of structural coherence.

## 5. Alternative Views

The proposal to incentivize latent exploration challenges the prevailing zero-tolerance orthodoxy regarding hallucinations. In this section, we engage with three primary credible positions that oppose our framework. These positions are grounded in safety engineering, optimization theory, and epistemology. We argue that while the risks of hallucination are real, the cost of their total elimination is the cessation of discovery.

### 5.1. The Safety Objection

A primary objection to incentivizing latent exploration concerns safety. Critics argue that any deviation from ground truth poses an unacceptable risk of misinformation (Weidinger et al., 2021). They contend that distinguishing useful exploration from factual error is computationally intractable in real-time. This view suggests that respecting hallucinations is inherently dangerous.

However, this binary classification of truth is insufficient for scientific discovery. We argue that global suppression of deviation induces a phenomenon we term Safety-Driven Mode Collapse. In this state, the model refuses to generate hypotheses in the long tail of the distribution. The danger lies not in the generation of novel hypotheses but in their presentation as verified facts. Our framework mitigates this via the Exploratory Signal-to-Noise Ratio metric. Unlike standard reinforcement learning from human feedback (Ouyang et al., 2022), which penalizes all divergence equally, this metric

acts as a conditional gate. It filters out structurally incoherent noise while routing coherent but unmapped outputs to a safety sandbox. This ensures that high-risk exploration is isolated from deployment without compromising the utility of the model as a hypothesis generator.

## 5.2. The Epistemic Limits of LLM Evaluators

In open-ended domains, an instructionally decoupled LLM-as-a-Judge can serve as a heuristic validator of structural coherence, but it should not be treated as an oracle. LLM evaluators are themselves trained models and may exhibit deterministic hallucinations, logical blind spots, or sycophantic alignment with the prompt. Thus, an LLM-informed ESNR score is only a screening signal, not proof of discovery.

Its role is to reject obvious Type I errors by checking internal coherence, self-consistency, and compatibility with stated constraints. It cannot establish the empirical truth of a Type II candidate. Definitive validation still requires external grounding through formal solvers, simulators, physical experiments, or delayed domain-specific verification.

## 5.3. The Stochasticity Objection

A second critique challenges the cognitive framing of the hypothesis. Skeptics argue that hallucinations are simply statistical artifacts arising from sampling noise. They suggest that framing them as cognitive exploration constitutes an anthropomorphic error (Shanahan, 2024).

While we avoid anthropomorphism, we draw a direct parallel to non-convex optimization theory. Deep neural network training relies on stochastic gradient descent and noise to escape local minima (Fotopoulos et al., 2024). Similarly, hallucinations during inference function as high-temperature sampling steps. These steps are necessary to traverse the energy landscape between disparate semantic modes. If a model is forced to strictly minimize perplexity against a static training set, the system freezes in the nearest local minimum of established knowledge. The deviations observed in generative discovery systems such as GNoME should not be dismissed as mere bugs by default. They can sometimes function as exploratory moves that shift the search from known solutions toward adjacent and previously unobserved regions of the state space. In this sense, controlled deviation can be useful for searching scientific hypothesis spaces.

## 5.4. The Decoder Reliability Objection

A further limitation concerns the decoder. HDCH describes exploration in latent space, but a latent traversal is useful only if the decoder can map the resulting representation into a coherent output. Since decoders are typically trained near the observed data manifold, their behavior in off-manifold

regions is not guaranteed. A geometrically meaningful latent path may still decode into an incoherent or misleading surface form.

Thus, discovery-oriented generation requires not only exploratory latent dynamics, but also robust decoding and post-generation validation. This motivates decoder-aware constraints, external validators, and safety sandboxes that test whether an off-manifold candidate remains coherent after projection into the output space.

## 5.5. The Epistemic Objection

A third objection questions the epistemic value of non-factual outputs. This position holds that an output contradicting established knowledge is false by definition and devoid of value.

We argue that this view conflates correspondence truth with coherence truth. Scientific history contains many theories that were factually wrong regarding the data of their time but structurally valid enough to guide future discovery. AlphaGo's Move 37 against Lee Sedol serves as a paradigmatic example (Sormani, 2023). The move had a near-zero probability in the human training distribution at the time of inference. A retrieval-based evaluator would have flagged it as an error. However, it possessed extreme structural coherence and strategic value. It was an innovation that expanded the boundaries of the game. Our framework provides the formal language to identify such moments in generative tasks. They appear as outliers in retrieval space but are optimal in latent utility space.

## 5.6. Falsifiability of the Framework

A final critique concerns the falsifiability of the Higher-Dimensional Cognitive Hypothesis. We propose that the hypothesis is empirically falsifiable. We suggest an experimental protocol comparing models optimized for high Latent-Traversal Scores against baseline models optimized solely for accuracy. The hypothesis is refuted if the exploratory models do not yield a statistically significant increase in valid scientific candidates compared to baselines. Results from systems such as GNoME are consistent with this possibility, but they do not by themselves establish the general claim. Systematic benchmarking is required to test this relationship across domains.

## 6. Mitigating Epistemic Mode Collapse

The suppression of hallucinations is motivated by safety, but it can introduce a systemic risk that we call Epistemic Mode Collapse. When a generative model is optimized strictly for high-likelihood or preference-aligned outputs, it may converge toward the mean of the training distribution and prune the long tail of low-probability but high-utility knowledge.

From this perspective, the ethics of hallucination is also a problem of distributional robustness: evaluation should reduce harmful confabulation without eliminating rare but potentially valid exploratory hypotheses.

## 6.1. The Statistical Cost of Zero-Hallucination Policies

Current alignment techniques often function as low-pass filters. They smooth the output distribution and reduce obvious confabulations, but may also suppress outlier reasoning that is statistically rare in the consensus data yet valid in specialized scientific contexts (Kandpal et al., 2023). Minimizing divergence from a human-preference distribution can also reduce model entropy and encourage sycophancy, where the model favors agreement with user assumptions over novel or contradictory truths (Perez et al., 2023). Categorizing all deviations as errors therefore trains models to avoid epistemic risk and limits their role to retrieval rather than discovery.

## 6.2. Protocol 1: Representation of Tail Distributions

Evaluation frameworks should value the representation of tail distributions. Many scientific insights reside in high-surprisal regions, where an output is unlikely under the training distribution but still structurally valid. We propose the Tail Coverage Ratio as a measure of the proportion of valid high-ESNR outputs generated from the bottom region of the probability mass. A robust model should maintain non-zero tail coverage, ensuring that minority scientific theories, non-standard solutions, and rare heuristics are not erased by majority-pattern aggregation.

## 6.3. Protocol 2: Temporal Validation Mechanisms

A core challenge in evaluating exploration is the temporal lag between hypothesis generation and verification. In dynamic semantic environments, such as public sentiment analysis, model outputs may require multiphase processing because the underlying social signal changes over time (Hao et al., 2023b). Similarly, a hallucination today may become a validated discovery tomorrow. We therefore advocate a Temporal Validation Protocol in which high-ESNR outputs receive provisional status rather than immediate binary judgment. This status can decay or strengthen over time based on external verification, such as wet-lab results, formal proofs, simulations, or delayed empirical evidence. The crystal predictions from GNoME illustrate this logic: their value was not determined by retrieval similarity, but by subsequent domain-specific validation.

## 6.4. Implications for Automated Science

The shift from error suppression to variance management is critical for automated science. Enforcing a zero-tolerance policy for hallucination precludes serendipity, where valuable insights emerge from unexpected exploratory paths. We propose the ESNR metric and Safety Sandboxes as controlled evaluation environments in which model temperature can be raised to induce exploratory reasoning while keeping candidate hypotheses isolated from deployment. In privacy-sensitive domains such as biometric recognition, decentralized heterogeneous-model systems also illustrate why exploratory or uncertain outputs should be routed through robust validation and trust mechanisms before deployment (Hao et al., 2025a). Under this view, generative AI should act not only as an archive of past knowledge, but also as an engine for generating candidates that can later be tested.

# 7. Conclusion

This paper challenges the binary classification of generative outputs into truth and error. We propose that specific hallucinations represent valid geodesic traversals in high-dimensional latent space. These traversals are often necessary for scientific discovery. Current evaluation metrics rely heavily on retrieval accuracy. We argue that this reliance induces epistemic mode collapse and stifles the exploration of unobserved hypotheses. We introduce the Higher-Dimensional Cognitive Hypothesis and the Exploratory Signal-to-Noise Ratio. These contributions provide a geometric framework for evaluation. They allow researchers to distinguish between Type I stochastic noise and Type II structural innovation. Our analysis suggests that discovery-oriented generation benefits from calibrated instability rather than the total suppression of deviation. Our controlled experiments illustrate that exploratory yield can peak near a critical transition regime. This finding necessitates a move away from zero-tolerance policies toward managed risk in hypothesis generation.

**Call to Action** We urge the machine learning community to implement three specific changes to realize this potential. First, the field must establish discovery benchmarks that prioritize novelty over simple retrieval. These benchmarks should reward models for generating valid solutions that are absent from the training data, utilizing dynamic verification environments rather than static datasets. Second, researchers should adopt thermodynamic reporting standards. This practice involves documenting the phase transitions of models to identify the optimal temperature windows for innovation, rather than reporting single-point accuracy metrics. Third, developers should implement safety sandbox architectures. These systems isolate high-risk exploratory reasoning from final deployment, allowing models to sample from high-temperature distributions without contaminating user-facing outputs. Implementing these steps will transform AI from a tool for reproduction into an engine for expanding the boundaries of human knowledge.

## Acknowledgements

This work was supported by the National Natural Science Foundation of China under Grants 62506016, 62277001 and U25A20446, and the Science and Technology Development Fund, Macao S.A.R (FDCT) 0028/2023/RIA1.

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
