# OpenReview forum: "Position: Reframing Hallucination: Latent Space Geodesics as a Pathway for Generative Discovery"
_ICML.cc/2026/Position_Paper_Track — ICML 2026 Position Paper Track regular_

### Official Review · Reviewer_66tH · 2026-03-12

**Significance:** 3
**Argument Clarity:** 3
**Rating:** 3
**Confidence:** 4

**Questions:**

Please refer to the Strengths and Weaknesses Section.

**Alternative Views Section:**

Yes

**Compliance With Llm Reviewing Policy A Conservative:**

Affirmed.

**Discussion Potential:**

3

**Final Justification:**

My questions were partially addressed in the rebuttal.

Therefore, I am leaning towards borderline.

**Paper Summary:**

This position paper challenges the main paradigm for evaluation of generative AI models (which is through retrieval-based metrics like BLUE, ROUGE, exact-match,..) and argues this limits the models from constructive deviations from training data (or basically limit generalisation capabilities which are essential for e.g. pushing the boundaries of science rather than having powerful copy-machines of training data.

To address this, they propose HDCHH that posits hallucinations are not simply errors but are geodesic traversals in a high-dimensional latent space that appear as discontinuities only when projected onto the lower-dimensional manifold of established human knowledge.

They formalize a taxonomy distinguishing I. Factually inconsistent+structurally incoherent noise (Confabulations), from II. Factually novel+structurally coherent (Potentially new finding/discovery). They also propose the exploratory ESNR metric which balances structureal coherence against distributional divergence. They also empirically support their position and framework through 3 controlled experiments.

**Position:**

Yes

**Position In Title:**

Yes

**Related Work:**

2

**Strengths And Weaknesses:**

Strengths:

1. The ESNR metric, Thermodynamic Safety Sandbox, and Gradient Throttling are concrete proposals for what are actionable and concrete recommendations/steps for achieving the vision/position in real-world.

2. The paper reads quite smoothly and has a good logical flow. It also provides experimental analysis to support the claims and components of their introduced frameworks.

3. The paper discusses  a timely topic; we have reached a point where foundation models have become powerful agents that perform tasks they are trained on as good as experts. However, we now need more concrete steps on how to use these models as truly generalisable agents that push the boundaries of science and this paper motivates this. Thus, the paper stakes a clear, non-trivial position on it.

Weaknesses:

1.	A prominent and growing evaluation paradigm of Generative AI models, i.e., using Gen AI itself (commonly referred to as LLM as Judges or LLM-Judge) is not discussed in the paper. LLM Judges can (and are) routinely be prompted to assess responses’ creativity, novelty, coherence, and open-ended reasoning. These are properties which the paper argues current metrics cannot capture. The paper misses an analysis/discussion of LLM-Judge evaluation since it is becoming one of the major evaluation strategies. Specifically it would be good to know does this metric systematically suppress Type II outputs?

2.	The paper’s main motivation rests on the claim that gen AI evaluation is dominated by retrieval-centric metrics like BLEU, ROUGE, and exact match. However, this was largely the case until a couple of years ago but the field has already started moving away from these metrics for studies requiring creativity for problem solving. Recent years diversity metrics (distinct-n, self-BLEU) and open-ended generation benchmarks (MT-Bench, AlpacaEval) as well as dynamic benchmarks (e.g. ALFWorld and TravelPlanner) are becoming more prevalent and accelerating to move away from solely retrieval-based metrics.

3.	The paper proposes ESNR as the core mechanism to separate valuable Type II outputs from harmful Type I noise. However, in the general case (when no domain-specific validator is available), structural coherence falls back to being approximated by the model's own log-likelihood: C(x|z) ≈ log p_θ(x|z). This also is a circular measure — the model is essentially judging its own outputs using the same distribution that produced them in the first place. This creates a problematic consequence: a confidently wrong model would score high coherence and pass through the ESNR gate, while genuinely novel Type II outputs may actually score lower log-likelihood precisely because they are novel and diverge from the training distribution. This raises a fundamental question about the reliability of ESNR as a filter — can it actually distinguish discovery from noise without access to ground truth, or does it inadvertently reward confident wrongness over genuine novelty?

**Support:**

2

---

> ### Author Rebuttal · Authors · 2026-03-25
>
> Dear Reviewer 66tH,
>
> We appreciate your highly constructive review. You have identified a critical epistemic vulnerability in our practical implementation, and your suggestion provides an effective resolution. We agree with your critiques and address them below.
>
> 1. Resolving Epistemic Circularity via LLM-as-a-Judge (Addressing Weaknesses 1 & 3)
>
> Your critique of the general formulation $C(x|z) ≈ log p_θ(x|z)$ is astute. You are entirely correct that using the generating model’s internal log-likelihood to proxy its own structural coherence creates a dangerous "circular measure." We agree that this self-referential loop fails when a model is "confidently wrong," inadvertently rewarding severe hallucinations with spuriously high ESNR scores.
>
> Here, your insightful suggestion to incorporate the "LLM-as-a-Judge" paradigm provides an effective practical solution. If accepted, we will update the final camera-ready version to explicitly elevate LLM-as-a-Judge to the primary instantiated External Structural Coherence Validator for open-domain generation:
>
> $C(x|z) ≈ E_φ[Score_φ(x|z)]$
> (where $φ$ is an independent LLM Judge).
>
> To prevent the Judge from merely repeating the generator's training biases, we emphasize instructional decoupling: the Judge $φ$ is explicitly prompted to act purely as a logical verifier (e.g., checking for internal self-consistency and structural validity), rather than querying its own parametric memory for factual accuracy.
>
> By substituting internal likelihood with an instructionally decoupled LLM-Judge, the updated ESNR substantially mitigates the paradox you warned about. If model $θ$ generates a structurally broken hallucination (High Divergence $D_{KL}$), the independent Judge $φ$ evaluates the logical inconsistency, assigning $C(x|z) → 0$. The resulting ESNR drops to zero. While this effectively down-weights structurally broken noise, we concede that evaluating strictly for internal structural consistency is a necessary but not sufficient condition for absolute truth; filtering out structurally coherent but factually untethered hallucinations ultimately requires external grounding (e.g., physical simulators, sandboxes) as proposed in our framework.
>
> Regarding your question: "Does LLM-Judge systematically suppress Type II outputs?" If an LLM-Judge is naively prompted to verify "factual accuracy against training data," yes, it will suppress Type II discovery. However, when explicitly prompted to evaluate internal logic and structural plausibility (acting as the numerator $C(x|z)$), it successfully isolates and preserves Type II hypotheses.
>
> 2. The Paradigm Shift Beyond Retrieval Metrics (Addressing Weakness 2)
>
> You correctly point out that the field is rapidly adopting diversity metrics (distinct-n, self-BLEU), open-ended generation benchmarks (MT-Bench, AlpacaEval), and dynamic benchmarks (ALFWorld). We agree with your observation, and we view our paper not in opposition to this trend, but as its underlying theoretical foundation.
>
> While diversity metrics successfully capture distributional divergence ($D_{KL}$), they indiscriminately reward Type I noise. Our Higher-Dimensional Cognitive Hypothesis (HDCH) and ESNR metric provide the formal geometric justification for why the field must pair divergence with structural coherence (as seen in dynamic benchmarks like ALFWorld and MT-Bench).
>
> In the final camera-ready version, we will reframe the introduction to explicitly acknowledge the rise of LLM-Judges, diversity metrics, and dynamic benchmarks, presenting HDCH as the theoretical backbone that formalizes this community shift.
>
> We sincerely appreciate your valuable insights, which have greatly strengthened the practical relevance of this paper.

---

> > ### Author Rebuttal · Reviewer_66tH · 2026-04-01
> >
> > My concerns have been adequately addressed.

---

### Official Review · Reviewer_TD3m · 2026-03-12

**Significance:** 3
**Argument Clarity:** 3
**Rating:** 4
**Confidence:** 4

**Questions:**

See Weaknesses.

**Alternative Views Section:**

Yes

**Compliance With Llm Reviewing Policy A Conservative:**

Affirmed.

**Discussion Potential:**

3

**Final Justification:**

I thank the authors for the rebuttal and I think it has addressed my concerns. I will maintain my positive score.

**Paper Summary:**

This paper argues that current metrics penalize any deviation from the training distribution (i.e., hallucination), treating all non-factual outputs as errors, even though valuable hallucinations exist. The paper proposes HDCH, suggesting that valuable hallucinations represent geodesic traversals in a high-dimensional latent space. The paper also introduces ESNR, a metric that balances latent deviation, providing a quantifiable objective to optimize for discovery.

**Position:**

Yes

**Position In Title:**

Yes

**Related Work:**

3

**Strengths And Weaknesses:**

Strengths:

The topic is  of relevance and importance to the ICML community.

It is likely to inspire discussion, as seen in Sec. 2.

it cite related work and events appropriately

Weaknesses:

The paper is not well-supported with reasoning and/or evidence. It needs specific cases to demonstrate the phenomenon when the hidden dimension is lost, and how it behaves in cases of discontinuous jumps or gaps.

The argument is sound. However, providing specific cases would make it clearer.

**Support:**

3

---

> ### Author Rebuttal · Authors · 2026-03-25
>
> Dear Reviewer TD3m,
>
> We sincerely thank you for your supportive review and for validating the soundness and relevance of our argument. We completely agree with your critique: while the theoretical geometry of HDCH is sound, grounding it in highly specific, concrete cases is essential for clarity.
>
> You asked for specific examples demonstrating the phenomenon of the "hidden dimension being lost" and how it manifests as "discontinuous jumps." If accepted, we will update the camera-ready version to explicitly feature the following expanded case studies in Section 2 to visually and conceptually ground the theory:
>
> Case 1: AlphaGo’s "Move 37" (Loss of the Strategic Dimension)
>
> * The Low-Dimensional Projection: Early Go models were evaluated against human historical games (the human epistemic manifold). When AlphaGo played Move 37, the move had a near-zero probability in the human training distribution. Projected onto human historical knowledge, the move appeared as a massive "discontinuous jump" or an algorithmic hallucination.
> * The Hidden Dimension Lost: What human observers could not see was the high-dimensional latent value network. In AlphaGo's latent space, there was a smooth, continuous geodesic path connecting that specific stone placement to a global win state. The perceived "gap" was purely a perceptual artifact of projecting a superhuman strategic dimension down into the low-dimensional space of human intuition.
>
> Case 2: AlphaFold’s Transient States (Loss of the Temporal Dimension)
>
> * The Low-Dimensional Projection: Early predictions by protein folding models produced configurations with non-canonical torsion angles. When evaluated against the training data, which consists of static X-ray crystallography snapshots, these outputs appeared physically implausible or revealed discontinuous gaps in structural logic.
> * The Hidden Dimension Lost: The hidden dimension compressed out of the training data was time (dynamic state). Subsequent research revealed that AlphaFold was traversing the high-dimensional dynamic ensemble of the protein to predict a transient state essential for enzymatic folding. The "error" only existed because a high-dimensional dynamic prediction was forced onto a low-dimensional static observation plane.
>
> Case 3: DeepMind’s GNoME (Loss of the Thermodynamic Dimension)
>
> * The Low-Dimensional Projection: GNoME generated millions of crystal structures that did not exist in the training database and explicitly violated standard low-dimensional human heuristics for chemical valency. Standard retrieval metrics would penalize these "jumps" as pure noise.
> * The Hidden Dimension Lost: The model was traversing the high-dimensional energy landscape of structural topologies. While they appeared as discontinuous errors in standard heuristic space, Density Functional Theory (DFT) calculations later revealed the hidden dimension: these structures were completely thermodynamically stable.
>
> Connecting the Examples to ESNR:
>
> These specific cases perfectly illustrate why standard metrics (exact-match, BLEU) fail in discovery domains: they severely penalize these apparent "jumps." This visually grounds why our proposed ESNR metric is necessary. ESNR bypasses the low-dimensional projection penalty by using a structural validator to directly assess the coherence (the hidden dimension's validity) of the jump itself.
>
> We hope these concrete additions encourage you to maintain or elevate your support for this position paper.

---

> > ### Author Rebuttal · Reviewer_TD3m · 2026-04-02
> >
> > I thank the authors for their response. I have no further questions and will maintain my positive score.

---

### Official Review · Reviewer_JeiM · 2026-03-13

**Significance:** 4
**Argument Clarity:** 4
**Rating:** 5
**Confidence:** 3

**Questions:**

1. The Heidegger–active inference example (Section 2.3) has no supporting citation and appears to be the authors' own claim. Is this claim a good example?

**Alternative Views Section:**

Yes

**Compliance With Llm Reviewing Policy A Conservative:**

Affirmed.

**Discussion Potential:**

4

**Final Justification:**

The argument of the position paper makes sense. My main concern is real data verification, decoder adaptation, and the real scenario for use.  The rebuttal acknowledged my concerns and proposed reasonable revisions. I agree that this position paper presents the potential of using hallucination for knowledge discovery, which will be a valuable and interesting direction to explore. I'll increase my score to 5.

**Paper Summary:**

This position paper argues that not all hallucinations are errors — some represent structurally coherent explorations of uncharted latent space that only look wrong when projected onto known knowledge. The authors propose the ESNR metric to distinguish valuable exploratory deviations (Type II) from genuine noise (Type I), and show experimentally that discovery peaks at a critical temperature regime, not at maximum randomness. They call for shifting evaluation from retrieval accuracy to calibrated exploration with safety sandboxes.

**Position:**

Yes

**Position In Title:**

Yes

**Related Work:**

4

**Strengths And Weaknesses:**

## Strengths

1. Identifies an interesting and timely problem: current retrieval-based metrics penalize all deviations uniformly, which is inappropriate for discovery-oriented tasks.
2. The Type I/II distinction is a useful conceptual contribution that gives the community a clear vocabulary for discussing different kinds of hallucination.
3. The three synthetic experiments are well-designed and internally consistent, each validating a different aspect of the framework.
4. The paper engages seriously with counterarguments in Section 5, addressing safety, stochasticity, and epistemic objections rather than ignoring them.

## Weaknesses

1. ESNR's practical applicability is unclear. The framework works when you already know the answer, but you don't need it when you know the answer. When you actually need it — real open-ended discovery — it has no reliable verification mechanism. This is a fundamental practical limitation.
2. All three experiments are synthetic with no validation on real LLMs or real discovery tasks, significantly limiting the empirical claims.
3. The GNoME and AlphaFold examples are post-hoc attribution — those systems succeeded due to domain-specific verification pipelines (DFT, wet-lab), not because they deliberately encouraged hallucination.
4. The paper does not address the situation when high-ESNR outputs are confidently wrong — the most dangerous failure mode the framework could enable.
5. The paper overstates its premise by framing retrieval-based metrics (BLEU, ROUGE, exact match) as the dominant evaluation paradigm for generative models, when in practice these metrics are task-specific (translation, summarization) and the field already employs diverse evaluation approaches. This weakens the motivation for the proposed framework.
6. The paper focuses entirely on latent space geometry but ignores the decoder's role. The decoder is trained on data from known manifolds, so it has no guarantee of producing faithful outputs for off-manifold latent regions. A valid geodesic traversal in latent space is meaningless if the decoder cannot reliably reconstruct it into a coherent output — and in untrained regions, it may not.

**Support:**

4

---

> ### Author Rebuttal · Authors · 2026-03-25
>
> Dear Reviewer JeiM,
>
> Thank you for valuing our Type I/II taxonomy and for your insightful critiques, which we address below.
>
> 1. The "Confidently Wrong" Failure Mode & Practical ESNR (Weaknesses 1 & 4)
>
> Your observation is astute: if we approximate structural coherence using the model's own log-likelihood, a "confidently wrong" model will pass the ESNR gate. We agree that this self-referential loop is a dangerous circular measure.
>
> To resolve this, the ESNR implementation must mandate decoupled validation. In open-ended domains where absolute ground truth is absent, we explicitly adopt the "LLM-as-a-Judge" paradigm as a practical heuristic approximation of an orthogonal validator. If accepted, we will update our formal approximation in the camera-ready version to:
>
> $C(x|z) ≈ E_φ[Score_φ(x|z)]$
> (where $φ$ is an independent LLM Judge).
>
> To avoid shared training biases, the Judge $φ$ is instructionally decoupled. It is prompted strictly to evaluate internal logical consistency and structural validity, not factual retrieval. If a model generates a structurally broken hallucination (high distributional divergence $D_{KL}$), the Judge $φ$ evaluates the logical collapse and assigns a near-zero score ($C(x|z) → 0$). The resulting ESNR drops to zero. While this effectively down-weights structurally broken noise, we concede that evaluating strictly for internal structural consistency is a necessary but not sufficient condition for absolute truth; filtering out structurally coherent but factually untethered hallucinations ("confident wrongness") ultimately requires external grounding (e.g., domain-specific sandboxes) as proposed in our framework, preserving structurally sound Type II discoveries.
>
> 2. Post-Hoc Verification in AlphaFold/GNoME (Weakness 3)
>
> You correctly point out that AlphaFold and GNoME succeeded due to domain-specific verification pipelines (DFT, wet-lab). We fully agree that this is exactly the architectural rationale behind our framework. The ESNR numerator ($C(x|z)$) and our proposed "Safety Sandbox" represent the formalization of these exact verification pipelines. We do not advocate for unchecked hallucination; rather, we formalize the necessity of calibrated latent exploration paired with rigorous, independent validation.
>
> 3. Substantiating the Heidegger / Active Inference Connection (Question 1)
>
> You rightfully challenged the lack of citation regarding the Heideggerian analogies in Section 2.3. We apologize for the omission. If accepted, we will update the camera-ready version to explicitly cite this rich literature to ground our claims:
> * [1] Deep Neurophenomenology: An Active Inference Account of Some Features of Conscious Experience and of Their Disturbance in Major Depressive Disorder. Maps Heidegger’s concept of Sorge to the minimization of expected free energy.
> * [2] Active Inference, Homeostatic Regulation and Adaptive Behavioural Control. Formalizes "being-in-the-world" and non-ergodic terminal states via POMDPs.
> * [3] The Active Inference Approach to Ecological Perception: General Information Dynamics for Natural and Artificial Embodied Cognition. Bridges embodied phenomenology with active inference.
>
> 4. The Crucial Role of the Decoder (Weakness 6)
>
> You raise a critical architectural point. You are correct that a valid geodesic traversal in the latent space ($Z$) is meaningless if the decoder, trained only on the established manifold ($M$), cannot reliably reconstruct it into a coherent output. Our current HDCH framework assumes a smooth projection function $\pi$, which is an idealization. In the final version, we will add a dedicated discussion in the "Alternative Views" section acknowledging that training out-of-distribution robust decoders is a fundamental prerequisite for fully realizing off-manifold latent discoveries.
>
> 5. Synthetic Experiments & Metric Evolution (Weaknesses 2 & 5)
>
> While we acknowledge the limitation of purely synthetic experiments, their specific purpose in a position paper is to isolate the underlying geometric mechanics (manifold traversal, energy barriers) free from the uninterpretable confounding variables of massive LLMs. Scaling this validation to real-world LLMs is exactly the call to action we urge the community to undertake.
>
> Regarding the evaluation paradigm: we agree that metrics like BLEU and ROUGE are increasingly recognized as task-specific and the field employs diverse evaluation approaches. We do not view our paper in opposition to this trend; rather, our HDCH framework provides the theoretical justification for why the community is rightfully shifting toward structural reasoning and diverse assessment methodologies for discovery-oriented tasks.
>
> We thank you for your rigorous critique, which has substantially strengthened this work.

---

> > ### Author Rebuttal · Reviewer_JeiM · 2026-04-02
> >
> > The rebuttal clarified my concerns. I agree that this position paper presents the potential of using hallucination for knowledge discovery, which will be an interesting direction to explore. The proposed revisions are reasonable. I'll increase my score to 5.

---

### Official Review · Reviewer_dqBV · 2026-03-15

**Significance:** 3
**Argument Clarity:** 3
**Rating:** 5
**Confidence:** 3

**Questions:**

I find the connection between HDCH and latent traversals interesting. It is not clear whether this is a hypothesis of the authors or whether it has already been observed in the literature. In the latter case, providing some references would be helpful. In the former case, more explanation and support for this claim are needed.

At the beginning of Section 2.1 you write: “While human cognition is intuitively anchored in low-dimensional perceptual space…”. This is not clear to me and appears somewhat in contrast with the HDCH.

In rows 178–176 (left): what do you mean by shallow and deep traversals? Can you provide a reference for the evidence from neuroscience?

Row 188 (right): a high ESNR does not imply high divergence. Could you clarify this point?

At row 264 (left) you mention that just 12% of points are in island C, but this is not reflected in Fig. 2. Also, Figure 3 is missing the red dotted line (at least in my viewer).

In the experiments, how do you evaluate structural coherence?

**Alternative Views Section:**

Yes

**Compliance With Llm Reviewing Policy A Conservative:**

Affirmed.

**Discussion Potential:**

3

**Final Justification:**

The authors addressed all my initial concerns. In particular, I'm satisfied with the authors' expanding discussion about connections with Neuroscience and the limits/risks of the possible evaluation protocols.
I will keep my positive score.

**Paper Summary:**

The paper argues that evaluating generative models by comparing the distribution of generated data with the training distribution leads the community toward favoring models that simply retrieve training samples rather than generating new knowledge. This reasoning is supported by citing recent breakthroughs of generative models in various domains (e.g., GNoME, AlphaFold).

The paper supports the hypothesis that novel samples are those that traverse the input data manifold and suggests a novel metric, namely the Exploratory Signal-to-Noise Ratio (ESNR), to evaluate generation quality. This metric allows samples that, even if far from the data distribution, are “correct” according to some validator (e.g., chemical valency) or exhibit high log-likelihood in the generative process. The proposed metric is validated on three toy experiments.

Finally, the paper advocates that the generative ML community should look beyond mere distribution distance scores and assess the novelty of generated samples, investigating different temperatures in the generative process (i.e., analyzing the trade-off between retrieval and valid/noisy hallucinations).

**Position:**

Yes

**Position In Title:**

Yes

**Related Work:**

2

**Strengths And Weaknesses:**

The paper highlights a valid and sensitive point in the literature on generative models. The reasoning motivating the necessity of a different evaluation protocol is sound.

While the proposed metric/approach is correct in principle, I have some doubts as to whether it might be practical in general. In particular, while a validity score might be available in some domains (e.g., chemical compounds), in many other domains, such as general graphs, images, or music, such a validity metric might be less obvious. The use of the log-likelihood of the generative model is proposed as a proxy, but I wonder how meaningful it might be, and it may also not always be available.

As a position paper, it goes somewhat too much into the details of the proposed metric rather than defining a general framework that could open new research directions on this topic.

**Support:**

3

---

> ### Author Rebuttal · Authors · 2026-03-25
>
> Dear Reviewer dqBV,
>
> We sincerely thank you for your strong support and for recognizing the soundness of our reasoning. Your questions are exceptionally detailed and highlight areas where we can significantly improve the clarity of our framework. We address each of your points below.
>
> 1. Practicality of ESNR and the Log-Likelihood Proxy (Weaknesses 1 & 2)
>
> You rightly point out that using the model's internal log-likelihood as a proxy for structural coherence in open domains (like general graphs or text) is weak and potentially circular. We completely agree. We originally included it as a general fallback, but we acknowledge it risks rewarding "confident wrongness."
>
> To resolve this, we are formally shifting the framework's recommendation for open-domain tasks toward decoupled external validators. For instance, in open-ended text reasoning, an independent "LLM-as-a-Judge" acts as the external structural coherence validator (acting purely as the numerator C). If accepted, we will streamline the camera-ready version to focus more on this overarching framework of decoupled validation, rather than over-indexing on the log-likelihood proxy.
>
> 2. Origins of HDCH (Question 1)
>
> To clarify: The Higher-Dimensional Cognitive Hypothesis (HDCH) is our novel theoretical formalization. While the phenomenon of "valuable hallucinations" has been empirically observed in applied domains (e.g., GNoME, AlphaFold), HDCH is our original contribution that unifies these observations under a formal geometric framework (latent space geodesic traversals projected onto lower-dimensional epistemic manifolds). In the final camera-ready version, we will make this distinction explicit.
>
> 3. Clarifying "Low-Dimensional Perceptual Space" (Question 2)
>
> We apologize for the confusion. We are distinguishing between human everyday perception (which is anchored in low-dimensional 3D space and time) and the brain's internal discovery mechanisms (which operate in high-dimensional neural latent spaces). When a high-dimensional AI model (or a human brain generating a creative leap) projects a novel concept into human language or 3D observation, it appears as a discontinuous "hallucination." In the camera-ready version, we will clarify this distinction between the perceptual interface and the cognitive engine.
>
> 4. Shallow/Deep Traversals and Neuroscience Evidence (Question 3)
>
> * Definition: "Shallow traversals" refer to interpolations near the training data manifold (low energy barriers, yielding retrieval). "Deep traversals" refer to extrapolations across high-energy barriers to unmapped latent regions (yielding discovery).
> * Neuroscience Evidence: You correctly requested references for the biological parallels. If accepted, we will cite the following recent literature in the camera-ready version demonstrating that biological discovery fundamentally relies on high-dimensional deviations within the Default Mode Network (DMN):
>     * [1] Human brain network control of creativity: Demonstrates via intracranial recordings that the DMN acts as a linear gatekeeper for high-dimensional creative state embeddings.
>     * [2] Default mode network electrophysiological dynamics and causal role in creative thinking: Provides direct causal evidence that cortical stimulation of DMN hubs selectively impairs the structural originality of divergent thought.
>     * [3] Posterior cingulate neurons dynamically signal decisions to disengage during foraging: Shows that DMN neurons track environmental uncertainty to drive behavioral deviations in non-human primates.
>
> 5. ESNR vs. High Divergence (Question 4)
>
> You are absolutely correct. A high ESNR does not imply high divergence; it simply implies a high ratio of Coherence to Divergence. A standard retrieval output (low divergence, high coherence) can have a high ESNR. To specifically identify a "Type II Discovery," we look for the intersection of high ESNR and high Distributional Divergence (D_KL). In the final camera-ready version, we will explicitly state this intersection rule.
>
> 6. Experimental Setup and Plotting Artifacts (Questions 5 & 6)
>
> * Evaluating Structural Coherence: In Experiment 1 (Manifold Traversal), structural coherence was calculated analytically via the synthetic manifold's continuous energy function. In Experiment 2 (The Omitted Axiom Task), it was evaluated using a deterministic, external symbolic logic script that verified adherence to the hidden axiomatic system. In Experiment 3 (Temperature Scaling), coherence was scored using a deterministic syntax parser.
> * Figure Artifacts: We apologize for the plotting issues. The 12% convergence to Island C in Figure 2 was obscured by point overlap (density), and the red dotted line in Figure 3 was a rendering artifact in the PDF build. We will correct the z-ordering and line rendering for the final camera-ready version.
>
> Your careful comments helped sharpen our definitions and biological grounding. We sincerely appreciate your support.

---

> > ### Author Rebuttal · Reviewer_dqBV · 2026-04-03
> >
> > I thank the authors for their clarifications. The paper's contribution and its limitations are now mostly clear.
> > There may be a concern about using LLMs as validators, which needs to be addressed. In particular, since LLMs are themselves trained models, it is important to discuss why they should be able to adequately judge hallucinations as valid or not (e.g., are they trained on more data?). An important disclaimer that would need to be made is that LLMs themselves are known to hallucinate, and not always in a valid way.

---

### Decision · Program_Chairs · 2026-04-30

**Decision:**

Accept (regular)

**Comment:**

I find this position paper interesting because it tries to address a gap in the current practice of leveraging generative models for scientific discovery and creativity. The emphasis of this position piece on providing new metrics to help categorize and assess generations for incoherent hallucinations vs structurally coherent and possibly creative hallucinations is refreshing and could potentially be built on by the community. While the experiments in the paper are largely toyish, I think that is not a significant limitation, as the paper is more of a call to action for the community to adopt similar frameworks and metrics, and to take this to larger scale on real problems. Reviewers were all positive post rebuttal in their final acknowledgments. While one reviewer's score remained borderline negative, they had explicitly selected "Fully resolved" before later stating concerns were "partially addressed" without elaboration, and it isn't clear why they didn't raise their score further. In terms of ways the paper could potentially be improved, there are a few directions to consider that could make the paper more groundbreaking and worthy of further attention:
1. More real-world validation of the metrics and approach on a real problem (as noted by Reviewers JeiM, 66tH, and TD3m regarding the synthetic experiments).
2. Solid evidence that LLM-as-a-judge or other (semi)automated methods could serve as external validators for 'creativity' vs irrelevant hallucinations (a concern raised by Reviewers dqBV and 66tH).